# Role of Crystalline Si and SiC Species in the Performance of Reduced Hybrid C/Si Gels as Anodes for Lithium-Ion Batteries

**DOI:** 10.3390/nano13030458

**Published:** 2023-01-23

**Authors:** Samantha L. Flores-López, Belén Lobato, Natalia Rey-Raap, Ignacio Cameán, Ana B. García, Ana Arenillas

**Affiliations:** Instituto de Ciencia y Tecnología del Carbono, INCAR-CSIC, Francisco Pintado Fe, 26, 33011 Oviedo, Spain

**Keywords:** sol-gel process, hybrid gels, magnesiothermal reduction, silicon, silicon carbide, anode, Li-ion batteries

## Abstract

In recent years, the research on lithium-ion batteries (LIBs) to improve their lifetime, efficiency and energy density has led to the use of silicon-based materials as a promising anode alternative to graphite. Specifically, crystalline silicon (*c*Si) and silicon carbide (SiC) obtained from deposition or reduction processes (e.g., magnesiothermal reduction) stand out for their electrochemical properties. However, the synthesis routes proposed until now have limitations that make them difficult to afford or operate on a large scale. For this reason, in this work, carbon-silicon (C-Si) hybrid materials synthesized through an efficient route are evaluated as the potential precursor for the obtention of both *c*Si and SiC species in a single material. The feasibility and influence of the magnesiothermal reduction process were evaluated, and materials with 10 wt.% of reduced Si and 10–26 wt.% of SiC were obtained. Both species play a role in the improvement of the performance of silicon-based materials as anodes in lithium-ion batteries. In comparison with materials obtained by the reduction of silica gels and composites, the reduced C-Si hybrid gels stand out thanks to the homogeneous distribution and stability of the species developed.

## 1. Introduction

The depletion of fossil fuel resources in combination with our current level of consumption has accelerated adverse environmental, health and social effects. Among the actions that seek to mitigate these consequences, the solvency of energy solutions is one of the most critical lines of action, not only in the adaptation to green procedures and renewable sources, but also through improvements in the infrastructure, which contribute to a more efficient utilization of energy. In this regard, innovation in devices for energy storage is challenging the current systems, following a constantly increasing demand and being their continuous development for a key and global strategy [1,2].

Specifically, lithium-ion batteries (LIBs) offer remarkable features, such as low weight, compact size and versatility. However, in order to meet the current demands, it is necessary to improve their energy density and rate performance [1]. To this end, current LIBs research is mainly focused on the electrode active materials, considered the key components for the storage device [2,3]. Different materials with capacities based on lithium-ion intercalation or lithium alloying reactions have been tested as anodes. Among them, the most successful are those involving carbon (C) materials and silicon (Si) nanostructures [4,5,6]. In regards to carbon materials, those having a graphitic structure show the best performance, with high reversibility of the lithium intercalation and electronic conductivity [3,7]. However, graphite has a limited theoretical capacity of 372 mAh g^−1^ and suffers from the generation of lithium dendrites. Nevertheless, nongraphitic carbons, such as gels obtained from resorcinol-formaldehyde (RF), have been investigated as anodes for LIBs. These carbon materials have attracted attention for this application because their physicochemical properties can be tailored to be used as models or optimized to improve electrochemical performances [8,9,10]. On the other hand, crystalline silicon (*c*Si) has great relevance due to its high theoretical specific capacity (3579 mAh g^−1^ based on the formation of the richest lithium phase, Li_15_Si_4_) and low working potential (~0.4 V vs. Li/Li^+^), but its quick degradation because of the huge volume change variation (~360% when the Li_4_Si phase is formed with the alloying and de-alloying of lithium ions), low electrical conductivity and unstable solid electrolyte interface (SEI), considerably reduces the life of the electrode (continuous capacity fading), and therefore, limits its practical application [6,11,12]. 

Among the strategies developed to solve the above problems, mixtures of both elements have been tested using different configurations, such as Si nanoparticles dispersed on carbon matrices [13,14,15], deposited on carbon-based nanotubes [5] or Si spheres coated with a carbon layer [16,17]. In all these cases, while carbon provides a stable structure that allows the transport of lithium without collapse, silicon provides capacity, enhancing the anode’s performance. In this regard, very interesting results have been reported concerning the use of silicon carbide (SiC). This silicon compound has shown good stability during cycling, and recent research has proved that it is able to react with lithium, forming the Li_4_C phase, which is a good ionic conductor with great chemical and mechanical stability. Due to these properties and their relationship with a stable solid electrolyte interphase (i.e., SEI), SiC is considered an adequate active material for the anode [18,19]. Based on this reaction, the estimated specific capacity for crystalline SiC is 2638 mAh g^−1^, although capacities ~1200 mAh g^−1^ have been reported for different SiC structures [20,21].

Therefore, since the electrochemical properties of *c*Si and SiC complement each other, the presence of SiC may improve the capacity retention as well as decrease the irreversible capacity (C_irr_) of the *c*Si anode [18,22,23]. For this reason, LIB anode materials combining both compounds have recently been developed. In this context, Tzeng et al. [24] prepared materials with SiC and carbon layers deposited on crystalline Si flakes, obtaining capacities of approximately 900 mAh g^−1^ after 150 cycles with a Coulombic efficiency of 90%, while Yang et al. [25] reported a more promising anode material with a “yolk-shell” configuration (i.e., *c*Si spheres covered by a SiC layer) and capacities of 1000 mAh g^−1^ after 400 cycles. However, reaching the advanced structure of such materials requires costly and time-consuming synthesis procedures. Although the electrospinning technique has been evaluated as a simple alternative to obtain these advanced nanostructures with similar electrochemical results (capacities of approximately 600–1000 mAh g^−1^ after 50–500 cycles), their application in full batteries and under real conditions is still in the development stage [26].

On the other hand, to avoid the high toxicity and elevated costs of chemical vapor deposition of silanes, *c*Si and SiC are usually obtained through reduction processes. The simplest way to obtain high-purity Si from SiO_2_ is by carbothermal reduction, where the melting point of silicon (i.e., 1414 °C) is reached to achieve the loss of oxygen. However, due to the extreme treatment conditions and the higher costs associated with it, alternative processes are required to obtain Si nanostructures with efficiency and to spread their application [27,28]. In this regard, the addition of molten compounds (e.g., Li, Ca, Al and Mg) as reducing agents allows a considerable reduction in the temperature required for the reduction process; moreover, metals stand out due to higher yields [27]. Therefore, among the alternatives, Mg is the preferred choice because of its efficient implementation, mild reaction conditions (i.e., low atmospheric pressure and low temperature) and higher yield based on its reaction mechanism [27,28,29,30]:(R1)$${\mathrm{SiO}}_{2}+2\mathrm{Mg}\;\to \mathrm{Si}+2\mathrm{MgO}$$

Reaction 1 is exergonic and exothermic, so it takes place easily over a wide range of temperatures (up to 1000 °C) with high stability of MgO against SiO_2_ [27,28]. The magnesiothermic reaction has been studied using different silicon sources, finding a great impact on the process variables involved.

Based on Reaction 1, the stoichiometric Mg/Si ratio is 2, and its variation affects the yield and composition of the reduced material. Limited quantities of Mg lead to a poor reduction, promoting the interaction of the MgO subproduct with the remanent SiO_2_ and generating undesirable compounds that are complicated to remove (i.e., Mg_2_SiO_4_) (Reaction 2). On the other hand, an excess of Mg results in the formation of Mg_2_Si (Reaction 3), which requires longer reaction times to develop Si species by subsequent reactions (Reaction 4) [28,29,31]. Then, values slightly above the stoichiometric Mg/Si ratio are recommended for higher reaction efficiency.
(R2)$${\mathrm{SiO}}_{2}+2\mathrm{MgO}\;\to {\mathrm{Mg}}_{2}{\mathrm{SiO}}_{4}$$
(R3)$${\mathrm{SiO}}_{2}+4\mathrm{Mg}\;\to {\mathrm{Mg}}_{2}\mathrm{Si}+2\mathrm{MgO}$$
(R4)$${\mathrm{SiO}}_{2}+{\mathrm{Mg}}_{2}\mathrm{Si}\;\to 2\mathrm{Si}+2\mathrm{MgO}$$

The effect of the heating rate during the reduction process is related to the reaction rate and has significant effects on textural properties. At approximately 650 °C, Mg melts, and the reduction reaction is favored. High heating rates in combination with the exothermic nature of the reaction caused overheating of the system, giving rise to the formation of big agglomerates and the Mg_2_SiO_4_ species. As a result, big crystals are formed, and the porous structure of the precursor material is considerably affected. On the other hand, slow heating rates allow a gradual and homogeneous interaction between reactants, promoting an effective reduction (through the formation of Mg_2_Si), forming crystals of smaller sizes and preserving the porous properties of the precursor material [28,32,33].

Finally, the role of temperature and residence time is closely related to the reaction route established by the two variables mentioned above (i.e., the Mg/Si molar ratio and the heating rate). The development of crystalline species and the reaction yield are favored by increasing the temperature, but lesser differences are appreciated in the range between 750 °C and 950 °C. On the other hand, longer residence times favor the total conversion to *c*Si, but sintering could occur [27,28,33].

In addition, the properties of the reduced materials prepared have been related to their electrochemical performance [34,35,36]. Therefore, information about these two issues (i.e., optimization of magnesiothermal reduction to obtain Si species and their electrochemical performance) may be already found in the literature. However, it is worth remarking that there is a lack of studies describing an effective raw material and procedure to obtain the proper combination of *c*Si and SiC to be used as anodes in LIBs. This opens a great opportunity for the applicability of C-Si hybrid materials as a competitive source material for effective LIBs anodes.

In this work, the magnesiothermal treatment applied to C-Si hybrid gels is evaluated as a promising route for the synthesis of materials containing *c*Si and SiC species. The C-Si hybrid gels used as precursors were synthesized following a methodology previously developed by the research team, which has some advantages in terms of processing time and cost [37]. In addition, and for comparison purposes, silica gels and composites (i.e., physical mixtures of carbon and silica gels) were also synthesized and subjected to the reduction process. The influence of the magnesiothermal process on the final composition, porosity and structure was analyzed. Finally, all the reduced materials (reduced hybrid, silica and composite gels) obtained were evaluated as potential anode materials for LIBs by prolonged galvanostatic cycling vs. Li/Li^+^ at constant current density. This study examines the role of the different electrochemically active silicon species, developed in sol-gel materials, in the performance of lithium-ion batteries, in terms of reversible and irreversible capacity and retention of capacity during cycling.

## 2. Experimental Section

### 2.1. Synthesis of Sol-Gel Materials

The synthesis of the precursor materials was based on the sol-gel process and involved microwave heating as a tool to make it faster [37,38]. The silica gel was obtained from hydrolysis-condensation of tetraethylorthosilicate (TEOS, 99% Chem-Lab, Zedelgem, Belgium) under basic conditions, using the molar ratios presented in Table 1 for the preparation of the precursor solution, and NH_3_(ac) 2 M (25% Acros Organics, Geel Belgium) as the booster (adjusted at pH 8) and aging solution. In this case, microwave radiation was used to heat the precursor solution at 60 °C for 3 h to complete the gelling, and subsequently at 30 °C for 2 h for the aging step. Drying was performed under ambient pressure at 30 °C to avoid the structure collapsing. 

On the other hand, in the composition of the C-Si hybrid gels, silica comes from TEOS, while the carbon contribution comes from pyrolysis of an organic gel synthesized from resorcinol and formaldehyde (R, 99% Indspec and F, 37% Merck with 10% of methanol as stabilizer). The mixture of all these reagents was performed using the following molar ratios: water/TEOS and EtOH/TEOS ratios were the same as for sample SG, R/F was fixed on the stoichiometric ratio and R/TEOS was varied to modify the silica content (50 and 85wt%). All values are shown in Table 1. To boost the reaction between silicon and organic species, 5 vol.% of the total amount of TEOS was replaced by [3-(2-aminoethylamino)propyl] trimethoxysilane (AEAPTMES, 97% Sigma Aldrich, Darmstadt, Germany). Due to the high silica content on the samples, the same gelling and aging conditions were used, while drying was performed at 60 °C. The C-Si hybrids obtained were carbonized under an inert atmosphere (N_2_) at 1000 °C for 2 h. For comparative purposes, a composite was prepared by the physical mixture of sample SG and a RF-carbon gel. The latter was synthesized following the procedure reported in previous studies using a R/F molar ratio of 0.5, a dilution ratio of 5.7 and a pH value of 5.0 [39]. The obtained materials are referred to as SG, for silica gel; HG/50 and HG/85, for C-Si hybrid gels with different silica content and CG for the composite. 

### 2.2. Magnesiothermal Reduction

To obtain the reduced materials, silica and C-Si hybrid gels were powdered and homogeneously mixed with Mg using a mortar. The quantity of Mg added to the mixture was calculated based on the reduction reaction between Mg and Si. For silica materials, Mg/Si values close to the stoichiometric ratio (i.e., 2) favored higher reaction efficiencies. Therefore, the Mg/Si ratio was set to 2.5. For C-Si hybrid gels, the reaction was not only influenced by the Mg added, but also by the silica content in the initial composition of the hybrid. Then, for the optimization of the magnesiothermal reaction in C-Si hybrids, two chemical variables were evaluated, the initial chemical composition (i.e., silica content of 50 and 85 wt.%) and Mg/Si ratio (i.e., 2.5 and 5).

In all cases, the mixture of Mg and silicon precursor was heated in a tubular furnace under an inert atmosphere (i.e., Ar, 300 mL min^−1^). Since the sol-gel materials synthesized for this study exhibit well-defined porous properties, three different heating rates were employed to evaluate their influence on them. An initial heating rate of 50 °C min^−1^ was used in all cases to reach 350 °C, and this temperature was maintained for 30 min. Afterwards, a second heating rate of either 1, 5 or 50 °C min^−1^ was used to reach 750 °C. This final set point was maintained for 12 h. The final treatment temperature and time of reaction were not varied in this work. The obtained samples were washed for 4 h under stirring with a hydroalcoholic solution of HCl 1M (HCl:H_2_O:EtOH molar rates 0.7:4.7:8.9) to eliminate subproducts MgO, Mg_2_Si, and Mg_2_SiO_4_ [6,40]. Finally, the reduced materials were dried at 120 °C. Figure 1 shows a schematic representation of the complete magnesiothermal process. These materials are identified by placing an “r” before their name to indicate that it is the reduced sample, followed by the heating rate used in the reduction process. For example, rHG/85-5, corresponds to the material obtained from the reduction of HG/85 (C-Si hybrid material with 85wt% of silicon) using a heating rate of 5 °C min^−1^. 

Moreover, the reference composite (prepared by mixing 85 wt.% of the silica gel with 15 wt.% of the RF-carbon gel) reduced with a Mg/Si ratio of 3.5 (since lower ratios lead to the formation of Mg_2_SiO_4_ species) and using a heating rate of 5 °C min^−1^, was denoted as rCG/85-5.

### 2.3. Physicochemical Characterization 

Sol-gel materials (SG, C-Si hybrid, C/Si composite) were characterized in terms of porous properties and chemical composition. Samples SG and C-Si hybrids were dried under vacuum for 8 h at 60 °C and 120 °C, respectively, prior to characterization. 

Adsorption-desorption N_2_ isotherms at 77 K were performed to obtain the volume of mesopores (V_meso_), micropores (V_micro_), and the external surface area (S_ext_) (Tristar II 3020 from Micromeritics, Norcross, USA), whilst mercury porosimetry was employed to determine the volume of macropores (V_macro_) (AutoPore IV from Micromeritics, Norcross, USA). The percentage of porosity (P) was calculated by complementing both techniques, and corroborated with the value obtained from the differences between helium and envelope density (see Appendix A). In addition, a macroscale analysis of their morphology was evaluated using a scanning electron microscope (SEM, Quanta FEG 650). In the case of SG, the chemical composition was theoretically calculated based on the silica composition, while for HG/85 and HG/50 the composition was calculated based on the elemental analysis and the determination of ash contents. 

The reduced materials were dried at 120 °C under vacuum for 8 h before their characterization. Crystalline species and chemical composition were evaluated by X-ray diffraction (XRD, D8 Advance diffractometer Bruker, Massachusetts, USA) using a CuKα X-ray source operated at 40 kV and 40 mA. X-ray photoelectron spectroscopy (XPS, SPECS with analyzer PHOIBOS 100) was also employed to determine the chemical composition and the different silicon compounds in the materials. Samples were analyzed using a scanning monochromate Al source and step energies of 30 eV for high-resolution spectra. 

### 2.4. Electrodes Preparation, Cell Assembly and Electrochemical Measurements

The reduced materials (i.e., silica gel, hybrid and composite, rSG, rHG/85 and rCG/85, respectively) were used as active materials for the preparation of the working electrodes. First, homogeneous slurries with 70 wt.% of active material, 20 wt.% of sodium carboxymethylcellulose binder (NaCMC, Sigma-Aldrich, MW ~700,000) and 10 wt.% of carbon black electric conductive additive (Super C65, Imerys, Paris, France) were prepared by mechanical stirring in an Overhead Stirrer Eurostar 20 (IKA, Staufen, Germany). For rHG/85 and rCG/85 materials, the following procedure was used: (i) a solution of 1 wt.% of NaCMC in distilled water was prepared at 3000 rpm for 30 min, (ii) C65 was added to the NaCMC solution while the stirring continued at 1000 rpm for 10 min, (iii) the corresponding material was incorporated to the dispersion and stirred at 4000 rpm for 1 h, and (iv) the slurry was tape-cast onto a copper foil of 25 µm thickness (Goodfellow, >99.99% purity) using a doctor blade (with a 250 µm gap) and a monitored/automatic film applicator with a perforated vacuum table heated at 50 °C (Elcometer 4340). For the preparation of slurries from rSG materials, a mixture of rSG and C65 powders were mixed in isopropanol at 70 °C under stirring (1 h) before the preparation of the slurry, with subsequent evaporation of the solvent (the complete information about this procedure is detailed in previous works [13,41]). This extra step was required to ensure a good incorporation of silicon with the carbonaceous material and eliminate step (ii) in the preparation of the slurry. In addition, since the rSG materials presented poor adhesion to Cu, the copper foil was first coated with a pre-layer of C65 and NaCMC (in the same proportions as on the electrode) using an aerograph, followed by material tape-casting under vacuum at 30 °C. The resulting electrode tapes were dried on a vacuum table at 80 °C for 1 h in all cases. Then, disc-electrodes of 12 mm in diameter were cut out of the tape using a manual punch, dried overnight at 120 °C in vacuum, and weighed to determine the active mass (ca. 1.2 mg). The mass loading of the electrodes was selected from preliminary studies that demonstrated a decrease in the electrode capacity attributed to diffusion limitations with an increase in mass loading (see Appendix A). Finally, the electrodes were stored in a glove box (MBraun, France) under Ar atmosphere, in which the concentrations of oxygen and water are below 0.1 ppm.

Two-electrode (working and counter) Swagelok-type cells were selected to perform the electrochemical measurements of the reduced materials, and they were assembled in the glove box. The prepared electrodes acted as the working electrode, while a Li-metal disk (Merck/Sigma Aldrich, 99.9% purity, 12 mm in diameter) served as the counter-electrode. The electrodes were separated by two micro-fiber glass disks (WHATMAN GF/A, 12 mm in diameter) impregnated with 150 µL of electrolyte (1 M LiPF_6_ salt in a mixture of ethylene carbonate (EC): diethyl carbonate (DEC), 1:1, *w*:*w*, with ~1–5 wt.% of vinylene carbonate (VC)). The initial potential of the cells was 3.0 ± 0.2 V vs. Li/Li^+^. Due to this study, the potential always refers to the redox pair Li/Li^+^, the term voltage is used instead of potential.

To evaluate the performance of the reduced materials as anodes for Li-ion batteries, galvanostatic cycling tests of the cells were conducted in a BioLogic Potenciostat (VMP2/Z). These experiments were carried out in the 0.003–2.1 V voltage range, at an electrical current density of 500 mA g^−1^ for up to 300 charge-discharge cycles. The graphite Timrex KS6, a commercial material applied as the anode of LIBs, was used for the preparation and cycling of reference electrodes following the same procedure.

## 3. Results and Discussion

### 3.1. Textural Properties and Composition of Sol-Gel Materials 

Figure 2 shows the isotherms of the silica and C-Si hybrid gels. As expected, silica gel shows a type IV isotherm, with a main pore size in the mesomacropore range and a high development of micropores. In comparison, the shape of the isotherms of the C-Si hybrid gels corresponds to type II, typically of macroporous materials [42]. This difference is attributed to the interaction between organic and inorganic species. In the C-Si hybrids, the polymerization occurs quickly, which leads to the formation of big nodules with a more open porous structure that avoids the collapse produced during the drying step. The small presence of micropores in C-Si hybrids, which varies between HG/50 and HG/85, can be attributed to the microporosity created in the organic part during carbonization. Thus, the C-Si hybrid with a higher percentage of carbon has higher micromesoporosity (Figure 2 and Table 2). Regardless of the different distribution of the pore size, all of them have a similar percentage of porosity (Table 2).

On the other hand, the variation in the silica content is appreciated in the chemical composition of the sol-gel materials (Table 2). The silicon (Si) content decreases from 46.7 wt.% in SG to 39.8 and 24.5 wt.% in HG/85 and HG/50, respectively. Oxygen follows in the same trend, decreasing with the silica content, while the presence of carbon in C-Si hybrid gels increases from 11.2 wt.% in HG/85 to 43.2 wt.% in HG/50. These differences in silica content are expected to influence the subsequent reduction treatment performed and the final Si species obtained, as analyzed in the next section. 

### 3.2. Magnesiothermal Reduction Process and Properties of the Reduced Silica Gel, C-Si Hybrid Gels and C/Si Composite

#### 3.2.1. Influence of Mg/Si Ratio 

The Mg/Si molar ratio has a great influence not only on the magnesiothermal reduction in silica gels but also on the reduction of C-Si hybrid gels. The XRD patterns of the reduced hybrid gels are shown in Figure 3. Regardless of the Mg/Si ratio, the reduced hybrids HG/85 (rHG/85-50 and rHG/85-50-Mg) exhibit peaks at 2θ of 28.4°, 47.4°, 56.2°, 69.1° and 76.4°, which correspond to the (111), (220), (311), (400) and (331) planes of *c*Si [16,30,43,44], and peaks at 35.6°, 60° and 71.8° which belong to the (111), (220) and (311) planes of SiC [40,43]. Therefore, these results confirm the presence of the cubic crystalline phase in both species (i.e., Si and SiC). In addition, a higher presence of carbon in the C-Si hybrid precursor (Table 2) promotes the formation of SiC during the reduction process, as in rHG/50-50, only this species (i.e., SiC) can be appreciated in the diffractogram. This is due to the higher availability of C and the consequent Si/C molar ratio of less than one (Table 2) that drives the reduction reaction towards the formation of SiC. 

These results were corroborated with XPS analysis. Figure 4 shows the deconvolution applied to the Si 2p spectra of the reduced materials. Regardless of the Mg/Si ratio, the reduced hybrids HG/85 (rHG/85-50 and rHG/85-50-Mg) show four main peaks at 104.7, 103.4, 101.4 and 99.8 eV (Figure 4a,b), attributed to SiO_2_, Si-O-C, SiC and Si-Si moieties, respectively [45,46] (dotted curves correspond to the splitting signal of Si species). However, differences in the concentration of each one were observed due to the Mg/Si ratio. The reduced material prepared with the lower ratio (i.e., rHG/85-5) shows lower contents of oxidated compounds and a higher concentration of the species of interest, SiC and Si (Table 3), which is related to a better reduction reaction efficiency. These results are in good agreement with the trends reported for the reduction of silica materials [27]. 

The presence of such species was also confirmed in the C 1s and O 1s spectra (see Appendix A). It is worth noting that the corresponding SiO_2_ peak is not found at its typical value of 103.5 eV because of the presence of C species (see XPS of H/85 in Appendix A); this behavior was already previously observed [44,46]. On the other hand, the rHG/50-50 material shows a very well-defined peak for SiC, absence of Si-Si bonds and low content of oxygenated moieties (i.e., SiO_2_ and Si-O-C) in the XPS spectrum (Figure 4c and Table 3), which indicates that an efficient reduction reaction with SiC as the main product occurs. Based on these results, it can be concluded that the most effective way to obtain SiC and *c*Si species in a single material is the use of C-Si hybrid gel precursors with a higher Si content (i.e., HG/85) and a Mg/Si ratio slightly above the stoichiometric one (i.e., 2.5). Therefore, the following studies were performed under these conditions.

#### 3.2.2. Influence of Heating Rate 

The effects of the heating rate variation were evaluated in the reduction of silica and C-Si hybrid gels, keeping the Mg/Si ratio at 2.5 to favor the reaction efficiency and avoid the formation of undesired subproducts. The influence of the heating rate on the formation of crystalline species was demonstrated by XRD. As expected, those samples prepared from SG only show the characteristic bands of *c*Si (Figure 5a), while the C-Si hybrid (i.e., HG/85) leads also to the formation of SiC species (Figure 5b). However, regardless of the base material (SG or HG/85), the intensity of the crystalline species decreases with the heating rate. This decrease can be explained based on the slow melting rate of Mg and the consequent gradual formation of crystals due to the reduction in accumulated heat [28,33]. In comparison with the reduced materials obtained at a heating rate of 50 °C min^−1^, the intensity of the crystalline species bands decreases by 30% and 70% for those reduced at 5 and 1 °C min^−1^, respectively (values calculated using XRD analysis with commercial graphite as reference, Appendix A). Furthermore, it can be observed in Figure 5b that, in the case of C-Si hybrid gels, the use of the slowest rate prevents the formation of crystalline SiC in the reduced material. 

The surface chemical composition of the reduced materials was determined by XPS. 

The spectra of rSG materials are presented in Figure 4f–h. Unlike hybrid reduced materials, the XPS spectra of these samples only exhibit the peaks related to Si-Si species and SiO_2_ (at 99.5 eV and 103.5 eV, respectively) [47,48]. As reported in Table 3, rSG-1 material contains the lowest percentage of nonreduced Si species (ca. SiO_X_/Si-O-C 54.6%) and 30.3 wt.% of Si-Si species. Based on the reaction mechanisms described for Si materials, this behavior can be attributed to the preferential formation of Mg_2_Si under slow heating rates in combination with long reaction times, which results in a maximum reduction to Si [32,33]. On the contrary, the rapid heating rate used to prepare rSG-50 favored the formation of Mg_2_SiO_4_, which prevents the further formation of Si-Si even at long reaction times [33]. As a result, a higher quantity of nonreduced Si was quantified in this material. On the other hand, Figure 4a,d,e show the XPS of reduced materials obtained from HG/85 at different heating rates. In this case, a higher percentage of nonreduced Si species was detected at the lowest heating rate, which may be attributed to the high stability of the species involving carbon (i.e., SiC and Si-O-C) and the thermodynamics of the reduction reactions in those hybrid gels. Then, these reduction conditions present low efficiency and, thus, a higher presence of partially reduced species (i.e., Si-O-C, see Figure 4e for rHG/85-1) with just 10 wt.% of each species of interest (rHG/85-1 in Table 3). By contrast, the reduction reactions for rHG/85-5 and rHG/85-50 materials present a good reduction yield, resulting in practically identical XPS spectra (Figure 4a,d) corresponding to 9 wt.% of Si-Si and 26 wt.% of SiC, with only 36 wt.% of silicon in nonreduced species (Table 3). In these materials, the variation in the heating rate mainly affects the SiC percentage, while the formation of Si-Si species remains almost constant. The reduction mechanisms of silica in a C-Si hybrid material have not been deeply studied, and due to the important role of SiC in energy storage applications [43], the differences shown in this work may be the guideline for a better utilization of these materials.

The influence of the heating rate during the reduction reaction also has a significant impact on the textural parameters of the resulting materials. The isotherms of the rSG and rHG/85 materials using the three different heating rates are shown in Figure 6a,b, respectively. Regardless of the textural properties of the precursor, all the reduced materials show type II isotherms. Therefore, it can be concluded that, in all of them, a microporous [10] structure was obtained [42]. These results make it clear that the reduction process has a great influence on the final porosity, even higher than the precursor material itself. Moreover, regardless of the precursor, there is an increase in nitrogen adsorption at low and high relative pressures by decreasing the heating rate. This means that slow heating favors the presence of small pore sizes due to the gradual development of crystals. In the case of silica gels, the gradual reaction helps to maintain the porous structure of the precursor and preserve mesopores, while the formation of aggregates at higher rates favors the creation of pores at approximately 200 nm [28,49]. For C-Si hybrid gels, small pores are formed during the reduction treatment, probably due to the elimination of undesirable subproducts (i.e., MgO), leading to pores of sizes comparable to the crystals formed. A similar phenomenon has been reported in the reduction of nonporous silica materials [30,50,51]. Figure 6c,d show the resultant trend in the textural properties based on the heating rate employed (i.e., external surface area, meso- and micropore volume). As for silica, the main difference between C-Si hybrids reduced at different heating rates is the mesopore volume and its correlated external surface area (S_ext_) increases at low heating rates. It is worth remarking that, although the reduction process greatly affects the chemical structure depending on the base material employed (silica or C-Si hybrid gels), its effect is not that significant on the final porosity.

Figure 7 shows SEM micrographs with high magnifications of SG reduced using different heating rates (Figure 7a–c). It can be observed that the small pores of rSG disappear through the formation of agglomerates at higher rates. However, the morphology of rSG-5 and HG/85-5 materials is comparable at low magnifications, as seen in Figure 7d,e.

A heating rate of 5 °C min^−1^ was chosen to reduce the composite CG/85 due to the good textural and structural properties of rHG/85-5 obtained at this heating rate. Appendix A shows the chemical and textural properties of the reduced composite (denominated rCG/85-5). The XRD pattern is similar to that of rHG/85-5, demonstrating the presence of both crystalline Si and SiC components. However, in terms of percentage content, the XPS spectra of rCG/85-5 indicate a diminution of both of them. In fact, the considerable decrease in the formation of SiC for the composite (from 26 to 10.9%) is mainly attributed to the absence of chemical bonds between silica and carbon in this kind of precursor (Table 3). By comparing the isotherms (Appendix A), the textural parameters of rCG/85-5 and rHG/85-5 can be considered analogous. 

In summary, after the reduction treatment of the sol-gel materials studied, it is possible to obtain materials containing the Si species of interest (i.e, *c*Si and SiC). In terms of the textural properties of the resultant material, small differences were observed between the reduced silica gels, C-Si hybrids and C/Si composites. However, the variation in the chemical composition of the precursors affects the final content of the different species produced. At least 10 wt.% of *c*Si was developed on all reduced materials. For the SG precursor, the absence of Si-C bonds leads to an increase in the formation of *c*Si to 14 or 30 wt.%, depending on the heating rate used during the reduction process. In the case of the SG reduced samples, Si in a matrix of silica was obtained, while for reduced C-Si hybrid gels, the formation of crystalline SiC could be produced, resulting in reduced materials combining Si and SiC crystals.

### 3.3. Electrochemical Performance 

The electrochemical performance was evaluated for the reduced gels obtained at different heating rates using the same Mg/Si ratio of 2.5 (i.e., rSG and rHG/85 series). Figure 8 shows the plots of the specific capacity and the Coulombic efficiency versus cycle number from the galvanostatic cycling experiments at a current density of 500 mA g^−1^. 

Due to the different chemical nature of the reduced materials tested and mentioned in the previous section, variations in the electrochemical performance are expected. As seen in Figure 8, for a given heating rate, the evolution of the capacity during cycling of rSG- and rHG/85-based electrode series is quite similar. The gels reduced at 1 °C min^−1^ showed a high initial discharge capacity, reaching values of ~1214 and 915 mAh g^−1^ for rSG-1 and rHG/85-1, respectively. This capacity is related to their high Si content (Table 3) and the considerable development of Si-O-C in rHG/85-1 (Figure 4), since both have been shown to be active species in LIBs [6,22,36,51]. In addition, the pathways available because of the mesoporous structure and external surface area of these reduced materials (Figure 6) facilitate the interaction between the active species and the Li^+^ ions. However, this fact also implies the deterioration of electrode Si species in the first cycle, leading to initial Coulombic efficiencies (CE) with values below 60% (Figure 8), and therefore, very high C_irr_ (c.a. 45%, Table 4) [6,8,10]. Although rHG/85-1 has a lower initial discharge capacity, by the 10th cycle, its performance already exceeds that of rSG-1, demonstrating a better cycling stability attributed to the presence of carbon species (i.e., Si-O-C and SiC) in the reduced hybrid [15,24]. On the other hand, despite the higher development of the Si and SiC crystalline species in the gels treated at 50 °C min^−1^ (Figure 5), the poor reduction yield leads to lower initial discharge capacities within their respective series (Figure 8, Table 4) [35]. The lower C_irr_ (16–25%, Table 4) of these reduced gels can be attributed to their low *S_ext_* and the successful buffering performed by SiO_x_ and SiC [24,25,51], not only providing a more stable and active structure but also avoiding a high initial degradation of the crystalline species by the regulation of Li^+^ diffusion [23,50]. However, in long cycling tests, the stability of SiO_x_ has been shown to be much lower than that of SiC [22]. Therefore, the discharge capacity recovery of rHG/85-50 electrode between cycles 10 and 200 is ~74% versus ~35% for the rSG-50 electrode (Table 4). This behavior extends to the entire series since, given a similar heating rate, rHG/85-based electrodes show a higher percentage of capacity recovery (R) during cycling than rSG-based electrodes (Figure 8, Table 4). Finally, the intermediate properties of rSG-5 and rHG/85-5 materials, which effectively combine moderate development of porosity and crystalline species (Figure 5 and Figure 6), appear to be the optimal combination. Thus, overall, they show the best electrochemical performance with capacities of ~350 mAh g^−1^ after 200 cycles, moderate capacity retention along cycling and irreversible capacity in the first cycle below 30% (Figure 8, Table 4). To evaluate the competitiveness of these materials, their performance was compared with that of the commercial graphite Timrex KS6 cycled under the same conditions, which, as expected, showed great stability up to 300 cycles but a lower capacity of ~230 mAh g^−1^ (Appendix A).

It is important to mention that the precoating used to improve adhesion in the case of rSG gels may also improve capacity. A comparative test was performed with the rHG/85-50 deposited on precoated Cu foil, and an increment of the final capacity was detected (Appendix A). This increase is attributed to the carbon black added in the precoat, which is commonly used as a conductive additive but has been shown to be electrochemically active toward LiPF_6_-based organic carbonate electrolytes [51]. Taking this effect into account, the differences in capacity between rSG and rHG/85 could be greater than those shown in Figure 8. 

Since the initial capacities of all the reduced materials are in good agreement with their *c*Si contents (Table 3 and Figure 8), it can be concluded that this species is especially active for charge storage, while carbonaceous species, particularly SiC, may improve the capacity retention with no detriment to the *c*Si activity. Thus, Coulombic efficiencies of approximately 98% were reached after the initial cycles and maintained through cycling (Figure 8). This confirms a homogeneous distribution of Si and C molecules in the hybrid precursor [37], which promotes the good arrangement between *c*Si and SiC without the necessity of engineered structures and tedious synthesis processes [25,52]. 

Based on the results discussed in this section, it is clear that the containing species and configuration of the reduced gels obtained from C-Si hybrid precursors are highly favorable for their use as active materials in Li-ion batteries. However, the role of the precursor material should be analyzed in depth. To this end, the reduced composite material rCG/85-5 (referred to in the previous sections) was also tested as an anode to compare the electrochemical behavior of the containing active Si species in a nonhybrid structure with that in the hybrid one already discussed. Figure 9 shows the plots of the specific capacity and the Coulombic efficiency versus cycle number from the galvanostatic cycling experiments at a current density of 500 mA g^−1^ for the rCG/85-5 composite, the rHG/85-5 hybrid gel and the rSG-5 silica gel. As can be seen in this figure, the reduced composite and the reduced silica gel have similar initial capacities at the beginning and end of 300 cycles. Even so, after the initial cycles and up to cycle 200, the capacity provided for the composite is higher due to the less abrupt drop shown for this material, which could be related to the presence of SiC and Si-O-C species created in the reduction process (Table 3). However, the similarity with the reduced silica gel at the end of the cycling suggests that these bonds are not stable, and they break down, freeing Si, which could start to interact with the lithium ions, leading to electrode degradation (capacity fading), as it happens from the beginning in the cycling of rSG with Si-containing species (Table 3 and Figure 9). On the other hand, the differences with the reduced hybrid gel make evident the great impact of the presence of Si-C and Si-O-C bonds in the precursor material and their relevant homogeneous distribution in comparison with composites obtained by the mixing method [17]. These bonds favor the development of higher quantities of SiC and more stable structures, leading to cycling profiles with good capacity retention (Table 4). The C-Si hybrid precursors show a clear advantage as regards the other materials studied in this work, which encourages further development and optimization for their use as anodes in LIBs.

## 4. Conclusions

C-Si hybrid gels synthesized with the sol-gel method are promising precursors for the development of materials containing both crystalline silicon (*c*Si) and silicon carbide (SiC) species after their magnesiothermal reduction. Among the process variables, these hybrids are affected in a similar way to silica materials: the Mg/Si ratio must be slightly higher than the stoichiometric value to promote the conversion rate, while an intermediate heating rate (i.e., 5 °C min^−1^) balances the development of the crystalline species and the mesoporosity of the final material. However, the amount of C and Si in the C-Si hybrids is essential for determining the final silica species after the reduction reaction. Using a hybrid with high silica content (i.e., 85 wt.%) and the optimized variables for the reduction process, it is possible to obtain a material with a good content of *c*Si (10 wt.%) and SiC (26 wt.%) species. Moreover, the electrochemical performance of this material as an anode in LIBs clearly reflects the active role of the crystalline species developed. The presence of *c*Si influences directly the initial capacity, while SiC plays an important role in the capacity retention. Furthermore, the activity of both species is maintained throughout cycling, which implies that the hybrid material structure has great stability. Although further work is necessary to optimize the *c*Si and SiC content ratios, the reduced materials prepared from C-Si hybrid gels, which can be mass-produced, appear as a good alternative for improving the performance of LIBs.

## Figures and Tables

**Figure 1 nanomaterials-13-00458-f001:**
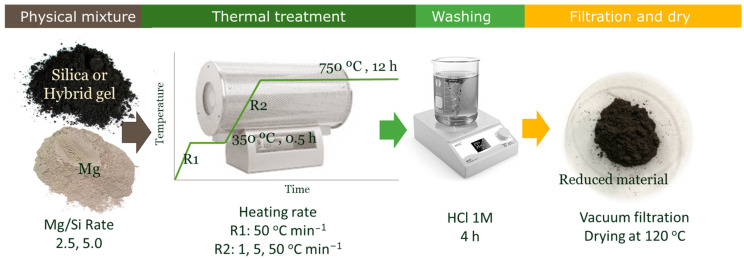
Schematic representation of the magnesiothermal process.

**Figure 2 nanomaterials-13-00458-f002:**
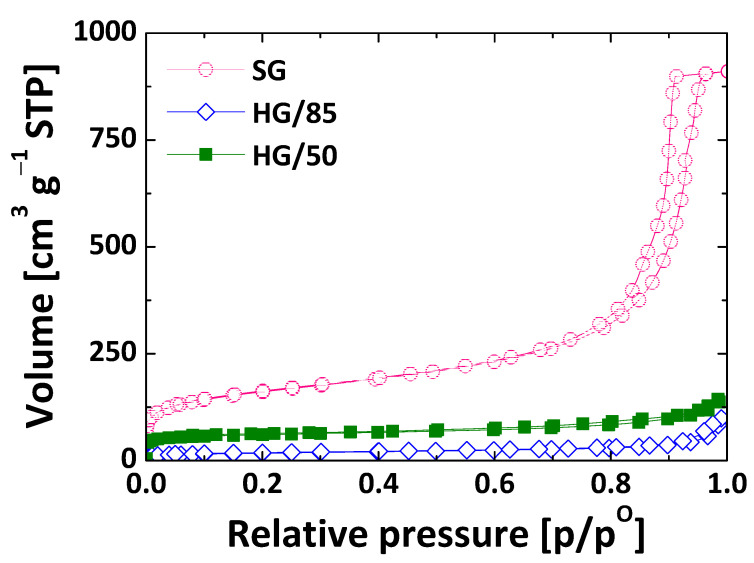
N_2_ adsorption-desorption isotherms at 77 K of sol-gel materials.

**Figure 3 nanomaterials-13-00458-f003:**
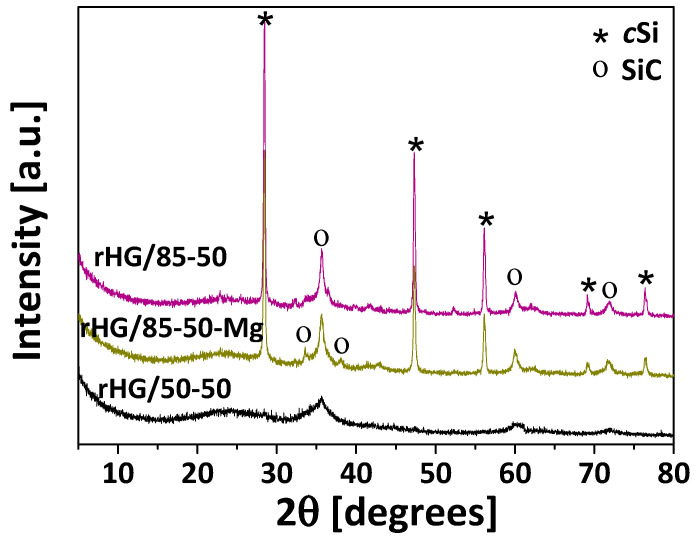
Crystalline species developed on reduced hybrid materials using different Mg/Si ratios.

**Figure 4 nanomaterials-13-00458-f004:**
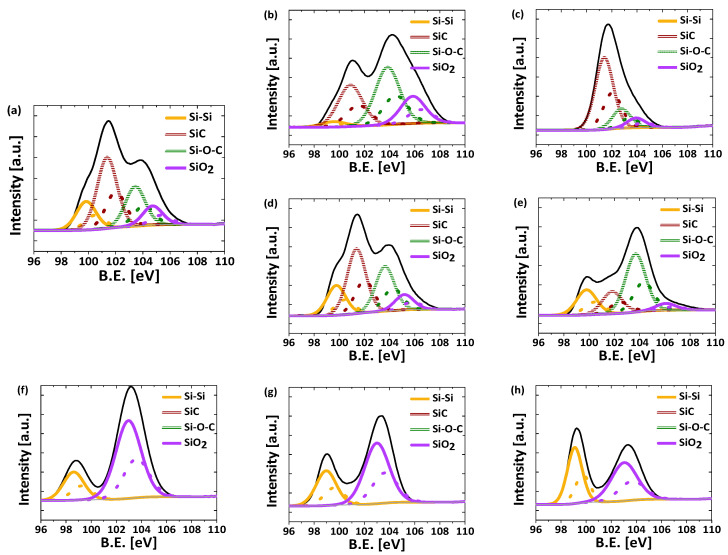
Si species obtained by the deconvolution of Si 2p XPS spectra for (**a**) rHG/85-50, (**b**) rHG/85-50-Mg, (**c**) rHG/50-50, (**d**) rHG/85-5, (**e**) rHG/85-1, (**f**) rSG-50, (**g**) rSG-5 and (**h**) rSG-1.

**Figure 5 nanomaterials-13-00458-f005:**
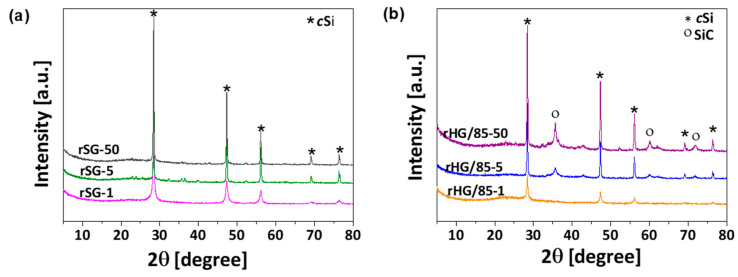
Crystalline species developed on reduced materials obtained from (**a**) silica gels and (**b**) C-Si hybrid gels precursors using different heating rates.

**Figure 6 nanomaterials-13-00458-f006:**
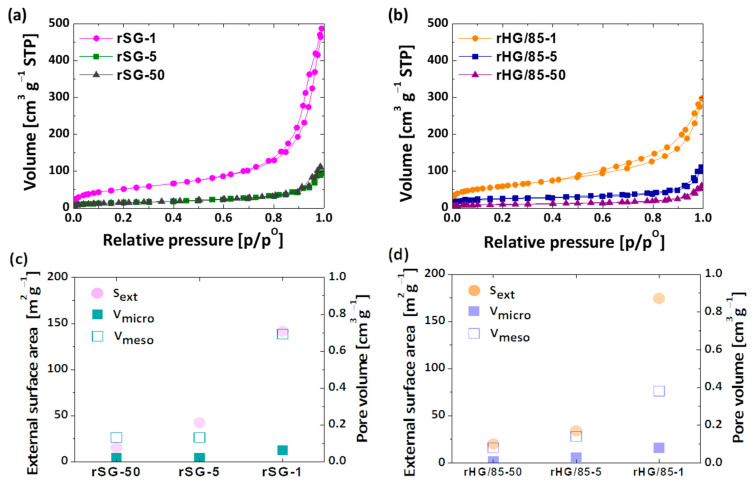
N_2_ adsorption-desorption isotherms and textural properties of reduced silica (**a**,**c**) and C-Si hybrid gels (**b**,**d**) using different heating rates.

**Figure 7 nanomaterials-13-00458-f007:**
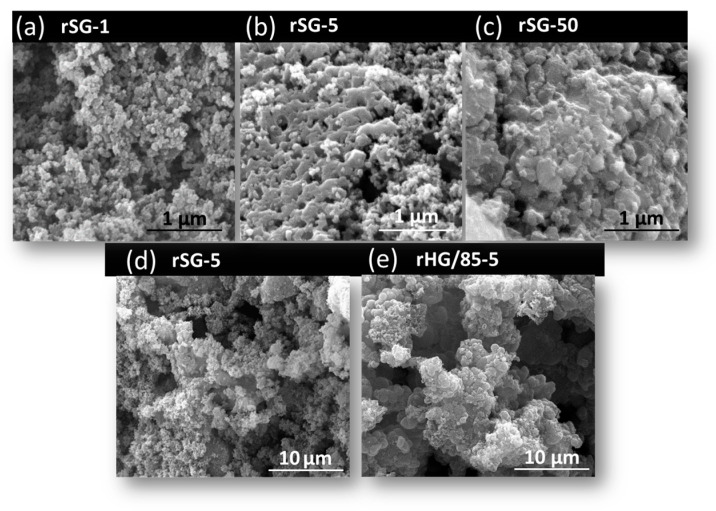
SEM images of reduced silica gels using different heating rates at high magnifications (**a**) rSG-1, (**b**) rSG-5, (**c**) rSG-50, and comparison between silica (**d**) and C-Si hybrid (**e**) gels at low magnifications.

**Figure 8 nanomaterials-13-00458-f008:**
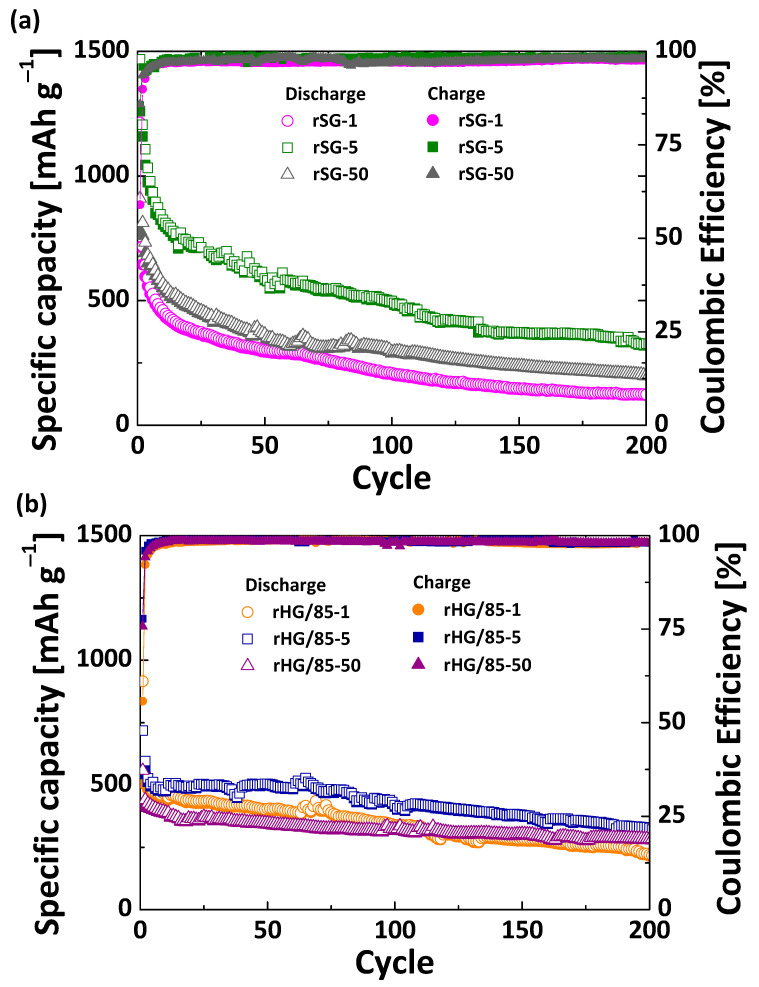
Specific capacity and Coulombic efficiency versus cycle number from the galvanostatic cycling at 500 mA g^−1^ of rSG- (**a**) and rHG/85-based (**b**) electrode series.

**Figure 9 nanomaterials-13-00458-f009:**
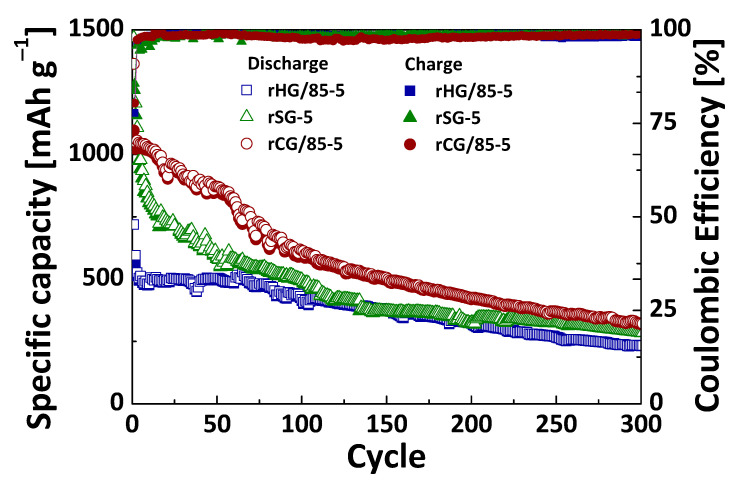
Galvanostatic cycling at 500 mA g^−1^ of silica, C-Si hybrid and C/Si composite reduced with a heating rate of 5 °C min^−1^.

**Table 1 nanomaterials-13-00458-t001:** Molar ratios employed in the synthesis of silica and C-Si hybrid gels.

Synthesis Variables				
Material	Water/TEOS	EtOH/TEOS	R/TEOS	R/F
SG	2.6	4.7	-	-
HG/85	2.6	4.7	0.1	0.5
HG/50	2.6	4.7	0.5	0.5

**Table 2 nanomaterials-13-00458-t002:** Porous properties and chemical composition of silica and C-Si hybrid gels.

Precursor Material	Porosity Parameters				Chemical Composition (wt%)				
	P (%)	V_micro_ (cm^3^ g^−1^)	V_meso_ (cm^3^ g^−1^)	V_macro_ (cm^3^ g^−1^)	C	O	H	Si	Si/C Ratio
SG	73	0.18	1.49	-	-	53.2	<0.1	46.7	-
HG/85	73	0.02	0.13	1.13	11.2	48.7	0.2	39.8	1.52
HG/50	83	0.10	0.17	1.40	43.2	31.7	0.6	24.5	0.24

**Table 3 nanomaterials-13-00458-t003:** Content of Si and SiC determined by XPS analysis, and the percentage of silicon remaining in nonreduced compounds in the materials studied.

Reduced Material	Composition (wt.%)		Non-Reduced Si(%)
	Si	SiC	SiO_x_/Si-O-C
rSG-50	14.4	-	78.9
rSG-5	20.2	-	68.9
rSG-1	30.3	-	54.6
rHG/50-50	0.2	40.0	28.6
rHG/85-50	8.6	26.9	36.4
rHG/85-50-Mg	2.7	20.0	64.1
rHG/85-5	9.2	26.0	35.9
rHG/85-1	10.7	10.0	64.4
rCG/85-5	6.8	10.9	64.4

**Table 4 nanomaterials-13-00458-t004:** Initial capacity, irreversible capacity in the 1st cycle (C_irr_, %) and capacity retention (R, %) obtained from the galvanostatic cycling at a current density of 500 mA g^−1^ of rSG- and rHG/85-based electrode series.

Material		*C_irr_* ^a^	Retention	
	Initial Capacity		*R_C10–100_* ^b^	*R_C10–200_* ^c^
	mAhg^−1^	%	%	%
rHG/85-50	558.5	25.2	83.9	74.3
rHG/85-5	718.7	24.4	83.6	62.2
rHG/85-1	915.4	44.2	72.3	46.9
rSG-50	909.5	15.6	51.8	35.3
rSG-5	1468.5	30.0	76.6	51.2
rSG-1	1214.9	46.9	45.7	26.6

^a^ Irreversible capacity C_irr_% = 100 (discharge capacity 1st cycle-charge capacity 1st cycle) (discharge capacity 1st cycle)^−1^. ^b^ Capacity retention *R_C10–100_*% = 100 (discharge capacity 100th cycle) (discharge capacity 10th cycle)^−1^. ^c^ Capacity retention *R_C10–200_*% = 100 (discharge capacity 200th cycle) (discharge capacity 10th cycle)^−1^.

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
