# Peer review of "Role of Crystalline Si and SiC Species in the Performance of Reduced Hybrid C/Si Gels as Anodes for Lithium-Ion Batteries"

_nanomaterials, 2023, doi:10.3390/nano13030458_

Round 1

Reviewer 1 Report

In this manuscript, Samantha L. Flores-López et al. reported carbon-silicon hybrid materials synthesized through the magnesiothermal reduction process and used as anodes in lithium-ion batteries. I would recommend this manuscript be accepted after addressing the following questions.

1. The core idea of the article need to be greatly simplified. For example, the title should emphasis the roles of crystalline silicon and silicon carbon.

2. The name of samples are chaotic. 

3. The authors should explain the reaction principle and advantage of magnesium thermal reduction more clearly.

4. In the conclusions, “seems to” is not recommended use in the article.

5. Some related references are suggested to be cited such as Science China Materials 2019, 62 (11), 1515-1536 and eScience 2021, 1, 141-162.

Author Response

Pleaser find attached the answers to your revision

Reviewer 2 Report

In this work, a carbon-silicon (C-Si) hybrid materials was synthesized through an efficient route, and the as-prepared sample exhibited good performances. The work is interesting, so the manuscript can be accepted after addressing the following comments.

1. The electrochemical performance of C-Si is closely related to the mass loading of electrode. The authors should do some research when the mass loading increases.

2.The SEI should be studied after cycling, such as the impedance. 

Author Response

Please find attached the answer to your revision

Round 2

Reviewer 2 Report

The manuscript has been revised and it can be accepted.